# Temporary Trends Concerning the Extent and Efficacy of Atrial Fibrillation Ablation Using Radiofrequency Energy in a Polish Single-Center Experience

**DOI:** 10.3390/medicina58020187

**Published:** 2022-01-26

**Authors:** Krzysztof Myrda, Piotr Buchta, Aleksandra Błachut, Michał Skrzypek, Mariusz Gąsior

**Affiliations:** 13rd Department of Cardiology, Silesian Center for Heart Diseases, 41-800 Zabrze, Poland; ola.blachut@gmail.com (A.B.); m.gasior@op.pl (M.G.); 2Silesian Center for Heart Diseases, 41-800 Zabrze, Poland; piotr.buchta@gmail.com; 3Department of Biostatistics, School of Health Sciences in Bytom, Medical University of Silesia, 40-055 Katowice, Poland; mskrzypek@sum.edu.pl; 43rd Department of Cardiology, Faculty of Medical Sciences in Zabrze, Medical University of Silesia, 40-055 Katowice, Poland

**Keywords:** atrial fibrillation, catheter ablation, efficacy, radiofrequency energy, trends

## Abstract

*Background and Objectives:* Atrial fibrillation (AF) is the most common supraventricular arrhythmia. Currently, catheter ablation is a preferred treatment strategy. The main objective of our study was a temporary trends analysis of patients’ data undergoing a single AF ablation procedure using radiofrequency energy (RF). The efficacy of the procedure underwent assessment during a 12-month follow-up. *Materials and Methods:* We analyzed 585 consecutive patients with symptomatic, recurrent, and drug-refractory AF hospitalized in our department between 2013 and 2018 who underwent RF ablation supported by a 3D electroanatomical system. The baseline characteristics, periprocedural parameters, and efficacy of the procedure at 6-, 9- and 12-month follow-ups were analyzed over the years. *Results:* The number of patients undergoing ablation increased. Patients with paroxysmal AF predominated (71.5%). However, the number of patients with the persistent type of arrhythmia increased over the years. The percentage of patients with chronic heart failure (CHF) increased to 27.5% in 2018, and patients presented with increasingly larger left atria (LA). In all patients, circumferential pulmonary vein isolation was performed. The percentage of patients who underwent arrhythmogenic substrate modification and cavotricuspid isthmus ablation increased. Over the years, the efficacy of a single procedure at the 12-month follow-up remained without significant differences between the years (72.0%, 69.6%, 75.5%, 74.8%, 71.7%, 71.7%). *Conclusions:* The rate of patients with CHF and advanced LA disease undergoing more extensive ablation increased over the years. The efficacy of a single procedure remained without significant differences between the years.

## 1. Introduction

Atrial fibrillation (AF) is the one of the most common heart rhythm disturbances, and its occurrence has a significant impact on higher healthcare costs, loss of productivity, more frequent hospitalizations, a higher risk of stroke [1,2], and increased mortality over the years [3,4]. Age is the leading risk factor for AF. In young patients (<49 years of age), this arrhythmia is rarely diagnosed and affects 0.1–0.2% of the population. The prevalence rate increases with age. In the population over 80 years of age, AF occurs in every fifth individual. It is also more prevalent in men (male to female ratio of 1.2:1) and in patients with comorbidities, including hypertension, heart failure, coronary artery disease, valvular heart disease, and cardiomyopathy [3,5,6,7]. The level of socioeconomic development of the country is also significant, with higher arrhythmia rates in developed countries. This may be influenced by differences in diet, prevalence of a sedentary lifestyle, and quantitative and qualitative differences in alcohol intake, which translates into a higher risk of metabolic syndromes and diabetes [8,9,10,11].

Catheter ablation is becoming an increasingly common treatment option for patients with AF [12,13,14,15]. In addition to the significant advantage of treatment in rhythm control strategies demonstrated in studies, we should also be encouraged by subsequent guidelines of cardiology societies, in which the class of recommendations for performing the procedure in patients with symptomatic AF is increasing. Attention is paid to the reduction in symptoms related to the arrythmia and its influence on a lower risk of stroke and better prognosis in the selected group of patients [16,17].

According to the current ESC recommendations, the main goal of the AF ablation procedure is to achieve the electrical isolation of PVs [17]. In a group of patients with paroxysmal AF without a structurally diseased heart and without an enlarged left atrium, the efficacy of pulmonary vein isolation (PVI) alone is high [18], and more than 80% of patients remain free of AF one year after the procedure. However, to maintain the sinus rhythm in selected cases, next to PVI, additional ablation lines, parasympathetic ganglion ablation, or an ablation strategy based on the recording of local atrial potentials are needed [19].

Unfortunately, information on the Polish population, representing the region of central and eastern Europe, eligible for such treatment, the adopted treatment strategy, and long-term outcomes is limited, which prompted us to perform the analysis of the data of patients who underwent catheter ablation in our department.

The main objective of this study was to analyze the temporary trends in the baseline characteristics and echocardiographic and periprocedural parameters, including the extent of a single AF ablation procedure using radiofrequency (RF) energy. The efficacy of the procedure underwent assessment during the 12-month follow-up.

## 2. Materials and Methods

### 2.1. Study Design and Study Population

This was a single-center retrospective cohort study evaluating not only the effectiveness and safety of RF ablation of AF but also temporary changes in the procedure and patient characteristics. We analyzed the data of all adult patients with symptomatic documented drug-resistant AF treated in our department between 2013 and 2018. According to the guidelines valid at that time, all patients underwent ablation with RF energy due to arrhythmia between 1 January 2013 and 31 December 2018.

Patients after previous surgical ablation or previous cardiac procedures with an incision in the atrial wall were excluded from the analysis. Cryoablation cases were not included in the analysis due to the lack of possibility of endomyocardial electroanatomic mapping performed, as per the standard procedure during each ablation procedure in our laboratory. The study flow chart for the analysis is shown in Figure 1.

Prior to any medical procedures, all patients signed a written informed consent form for invasive diagnostic procedures and treatment. The registry met the conditions of the Declaration of Helsinki and was approved by Bioethics Committee. 

### 2.2. Pre-Ablation Procedure

On admission, all patients underwent clinical and laboratory examinations, transthoracic echocardiography (TTE), and transesophageal echocardiography (TEE). Computed tomography (CT) for 3D cardiac imaging was routinely performed before the procedures in outpatient care settings. In the absence of contraindications, antiarrhythmic medication, including beta-blockers, propafenone, sotalol, and amiodarone, was continued during the hospital stay and at discharge. In-hospital anticoagulant treatment, including vitamin K antagonists (VKAs), such as warfarin; acenocoumarol; or novel anticoagulant drugs (NOACs), such as dabigatran, rivaroxaban, or apixaban, were administered according to the guidelines valid at that time. In cases of preprocedural high INR on admission, ablation was postponed until the next day(s).

### 2.3. Catheter Ablation and Post-Ablation Hospital Management

All procedures were performed by two EHRA-certified electrophysiology specialists (ECES). After obtaining written informed consent, the procedures were performed, and patients stayed under anesthesiological care during the whole procedure. Depending on clinical needs, anesthesia drugs, including sevoflurane and/or propofol with boluses of midazolam and fentanyl, were administered. The method of anesthesia delivery depended on the decision of the attending anesthesiologist. For invasive blood pressure monitoring, the routinely radial approach was used. Intraesophageal temperature measurement during ablation was achieved by a temperature probe (SensiTherm, St. Jude Medical Inc.; St. Paul, MN, USA). After femoral vein puncture under fluoroscopic guidance, decapolar steerable and quadripolar non-steerable catheters were placed in the coronary sinus and the right ventricular apex, respectively. A single transseptal puncture (TSP) was performed directly using a steerable sheath (Agillis, St. Jude Medical). After TSP, heparin was added to achieve activated clotting time (ACT) levels of 300–350 s. All ablation procedures were performed with the support of the electroanatomical 3D system (CARTO 3, Biosense Webster, Diamond Bar, CA, USA) or with the Ensite Velocity system (St. Jude Medical Inc.; St. Paul, MN, USA). The electroanatomical map of the left atrium (LA) was created and merged with reconstructed CT scans. In all cases, circumferential ablation around the pulmonary vein ostia was performed. Next, an electroanatomical map of the LA during sinus rhythm with a circular mapping catheter (Lasso, Biosense Webster, Diamond Bar, CA or Optima, St. Jude Medical Inc.; St. Paul, MN, USA) was created. In accordance with previous studies [20,21], peak-to-peak bipolar electrograms amplitude was defined as follows: normal if the local voltage was >0.5 mV, diseased ranged from 0.2 mV to 0.5 mV, and likely scar tissue with local voltage <0.2 mV. Low-voltage areas were defined as sites of ≥3 adjacent low-voltage points <0.5 mV and were modified with standard lines according to the localization: on the posterior wall—the line(s) between the upper and lower pulmonary veins, including roof line; on the septum—the line from the right superior pulmonary vein to the mitral annulus; and on the mitral isthmus—the line between the left inferior pulmonary vein and the mitral annulus (Figure 2). The typical flutter induced during the study or documented beforehand was treated with a cavotricuspid isthmus (CTI) line. The acute success of the procedure was defined as the bidirectional conduction block for all PVs lasting more than 20 min after the last application proven by both pacing along the ablation line with the ablation catheter and with a multipolar circular mapping catheter. When additional application lines were performed, their conduction block was confirmed by additional stimulation maneuvers. In the case of re-ablation, after evidence of reconnections between the LA and PV with a circular multipolar diagnostic catheter, re-isolation was performed first. For ablation, an irrigated catheter (ThermoCool, Biosense Webster or IBI Therapy Cool Flex, St. Jude Medical, Inc.; St. Paul, MN, USA) was chosen. The initial RF generator setting consisted of an upper catheter tip temperature of 43 °C, a maximal RF power of 35 W, and an irrigation flow rate of 30 mL/min. In RF, application was to the posterior wall, and the initial RF generator setting consisted of a maximal RF power of 20 W. The level of the temperature probe in the esophagus in reference to the ablation catheter was controlled using fluoroscopy. The maximum power delivered at the posterior wall and near the esophagus was reduced and adapted according to the intraesophageal temperature. To avoid an atrioesophageal fistula, the maximum acceptable intraesophageal temperature was limited to 41 °C [22]. Endoscopy of the upper gastrointestinal tract was performed in each patient with a temperature increase above 41 °C in the esophagus during RF application. Postoperative treatment was dependent on the recommendations of the consulting gastrologist. Regardless of esophageal temperature during ablation, patients were peri-procedurally started on proton pump inhibitors and continued for four weeks. After ablation and the removal of the sheath from the LA, protamine was administered to reverse heparin action. If the activated clotting time was lower than 200 s, the femoral sheaths were removed. Anticoagulation was continued 6 h after the procedure. We recommend continuing the previous anticoagulation and antiarrhythmic therapy after the procedure for at least 3 months. If complications occurred, we classified them as minor or major. Major complications were defined as transient ischemic attack (TIA), stroke, bleeding requiring blood transfusion, phrenic nerve palsy, pericardial tamponade, and arteriovenous fistula on the punction side. Minor complications were defined as local hematoma treated conservatively, pseudoaneurysm on the punction side, and pericardial effusion.

### 2.4. Procedure after Discharge

Patients discharged after ablation remained in outpatient care. There were three follow-up visits with ECG-Holter monitoring. Recurrence of arrhythmia was defined as the occurrence of a sustained (>30 s) episode of AF, atrial flutter (AFl), or atrial tachycardia (AT) documented in the electrocardiographic recordings due to the presence of subjective symptoms and documented rhythm disturbances in Holter-ECG recordings or in cardiac implantable electronic devices (CIEDs) interrogation. The period of the first 3 months was blinded. The decision to continue anticoagulation treatment in the follow-up depended on the clinical guidelines and individual risk reassessment.

### 2.5. Statistical Analysis

Continuous parameters with a normal distribution were presented as arithmetic means ± standard deviations, while qualitative parameters were presented as percentages. Normality of distribution was verified using the Shapiro–Wilk test. The parameter distribution, which was different than normal, was presented as a median with an interquartile range (IQR). Temporal trends were analyzed using the Jonckheere–Terpstra test for continuous data and the Cochran–Armitage test for categorical data. The relationship between the absence of arrhythmias in subsequent years of the follow-up was presented by means of the Kaplan–Meier curve, and the significance of differences was estimated using the logrank test. Statistical significance was set at *p* < 0.05. Statistics were calculated with SAS software, version 9.4 (SAS Institute Inc.; Cary, NC, USA).

## 3. Results

The group we analyzed included 585 patients, mostly with paroxysmal AF. This population was predominantly male (66%) and overweight (mean age 59 years). Baseline patient characteristics are given in Table 1. Over a period of 6 years, there were differences in the increasing risk of ischemic events according to the CHADS-VASc score: a higher percentage of patients with CHF and persistent AF (Figure 3).

Table 2 and Figure 4 summarize periprocedural data, including the extent of the procedures. Ablation limited only to PVI decreased in subsequent years. At the same time, the percentage of patients who underwent CTI ablation or arrhythmogenic substrate modification increased.

Freedom from AF/AFl/AT was maintained in 72.6% of patients at 12 months after the procedure. In the compared time periods, the efficacy remained comparable in the subsequent years (Table 3, Figure 5).

## 4. Discussion

The main observations over the years in our department are as follows: (1) The proportion of patients with CAF, CHF, and advanced LA disease undergoing AF ablation with RF energy significantly increased; (2) Despite this fact and the increasing necessity of substrate modification, the efficacy of a single procedure remained without significant differences between the years; (3) NOACs became the main antithrombotic drugs in the post-ablation period.

The analyzed population included predominantly male and overweight patients (mean age 59 years) with hypertension (67%), coronary artery disease (28%), diabetes (24%), and CHF (19%). This population was also predominant in the multicenter analysis from a Polish survey study [23]. In the registry, including populations from western and central European countries, the findings are similar. Patients undergoing ablation are younger and more symptomatic and have fewer risk factors. They are mostly diagnosed with hypertension, and the estimated risk of stroke and bleeding is lower [24]. The potential causes of the above findings are related to the fact that patients with paroxysmal arrhythmia are more symptomatic, and ablation in patients with fewer comorbidities is associated with a lower risk of perioperative events. Younger patients are often more aware of their treatment and are not willing to take more drugs or higher doses of antiarrhythmic drugs on a regular basis. Nevertheless, studies have suggested that ablation of AF may be a safe and effective procedure. It can also improve the prognosis in the group of older patients and those with more comorbidities [19]. One of the studies is CASTLE-AF [25], in which ablation in patients with heart failure was associated with a lower risk of death and hospitalization compared to drug treatment. This trend was also observed in our study. Over a period of 6 years, there were differences in the increasing risk of ischemic events according to CHADS-VASc score (Table 1) and a higher percentage of patients with CHF. Moreover, there was an increase in the proportion of patients referred for ablation due to persistent AF (Figure 3), in whom the LA dimension was significantly larger and who had lower LVEF. The causes for these changes in baseline characteristics could be attributed to the significantly increasing population of patients with heart failure in Poland [12,25] or even in the characteristics of the center from which the study comes. We are one of the largest cardiology centers in Poland that provide multi-profile care in this field, including qualification of patients for heart transplantation.

The change in the baseline characteristics of patients eligible for ablation coexisted with changes in the extent of ablations. The extent of the procedure involving only PVI (73% of all patients) decreased in subsequent years and reached 65% in 2018. At the same time, the percentage of patients who underwent CTI ablation and arrhythmogenic substrate modification increased from 17.3% in 2013 to 35% in 2018 (*p* = 0.0007). As a result, 70–75% of patients remained free of AF/AFl/AT 12 months after RF ablation in subsequent years of follow-up (Figure 4). However, it should be noted that due to the adopted methodology, the reported successful rate is likely to be lower. This trend remained consistent with the observations from other European countries [19,26,27,28]. The results of randomized trials, such as the SARA study [29], in which additional RF ablation lines were performed in more than 25% of patients, may support the validity of such an ablation strategy. It can also be confirmed by the analysis of patients with CHF included in the CABANA trial, in which additional ablation lines, parasympathetic ganglion ablation, or an ablation strategy based on the recording of local atrial potentials were allowed [19]. The high efficacy of hybrid ablation with minimally invasive access in the maintenance of the sinus rhythm could also support this approach [30,31].

Despite the invasive nature of ablation, it is a safe procedure. The percentage of serious complications is low, and they are usually related to the vascular access. It can be confirmed by the results of our analysis, in which minor complications (hematoma treated conservatively, pseudoaneurysm at the puncture side, or pericardial effusion) occurred in 3.9% of patients, and major complications (e.g., TIA, stroke, bleeding requiring blood transfusion, phrenic nerve palsy, pericardial tamponade, or arteriovenous fistula at the puncture side) affected 1.2% of patients. Importantly, the frequency of complications did not change over the years. Similar observations were made among 1000 patients treated with RF ablation in a high-volume center [32], where the percentage of all complications reached 3.9%, which were mostly related to the vascular access. In the analysis of the Polish population [23], major complications, such as tamponade or periprocedural stroke, affected 1.0% and 0.3% of patients undergoing ablation, respectively. It was consistent with the observations on the incidence of tamponade and perioperative stroke in the large registry of Capatto et al. [33], including 45,115 procedures between 2003 and 2006.

X-ray exposure at the time of cardiac procedures, during which fluoroscopy is the leading method of navigation, is also of significance. In the electrophysiology laboratory, the introduction of 3D electroanatomical navigation systems into daily practice had a significant impact on reducing X-ray exposure [34]. All patients included in this analysis underwent ablation using the above system. However, the search for alternative methods leading to the elimination of fluoroscopy as the standard management is a current challenge due to the serial occurrence of subsequent procedures with cumulative exposure of medical staff [35].

Over the years, we observed a significant change in the profile of medications used for antithrombotic prophylaxis (*p* < 0.0001). In 2013, at discharge from our department after ablation, NOACs were used in 25.3% of patients (Table 2). During the comparable period in patients from the European population, the frequency of NOAC use was also low. The use of VKAs was predominant also in western European countries [36], which can be confirmed by the data from the prospective GLORIA-AF registry from 2011 to 2013 [37] or the EORP-AF Pilot registry [38]. The change occurred in the following years due to subsequent reports, which was reflected in the 2016 guidelines [16] and the 2017 expert position statement [39]. Therapy with VKAs and NOACs was considered equivalent, with NOACs being the preferred drugs for the inclusion of de novo anticoagulation therapy. It contributed to a change in the profile of drugs used for anticoagulant prophylaxis, including observations from our center where NOACs were used by 74.2% of patients after ablation in 2018. A similar trend was also noted in the 2017–2018 analysis of other European populations [40], as well as in the population of Central Europe, which is represented by the Poles. In 2019, in a large Polish registry, 84% of patients with AF were treated with NOACs [41]. Confirming the validity of the findings from the registries and subsequent randomized data, the 2020 recommendations for the management of patients with AF unequivocally indicate that NOACs should be the preferred group of drugs for antithrombotic prophylaxis [17].

### Study Limitations

Our analysis was a retrospective, single-center study with the typical limitations of this method. The sample size was relatively small. The patients were recruited in a very long period of over 5 years. Only selected periprocedural and in-hospital data are available. The fluoroscopic data were not noted, and two different 3D electroanatomical systems were used. Efficacy assessments were based on clinical symptoms and limited data coming from Holter-ECG and CIEDs recordings.

## 5. Conclusions

The proportion of patients with CHF and advanced LA disease undergoing AF ablation with RF energy increased over the years in our department. Despite this and the necessity for more extensive ablation, the efficacy of a single procedure remained without significant differences over the years. NOACs became the main antithrombotic drugs in the post-ablation period.

## Figures and Tables

**Figure 1 medicina-58-00187-f001:**
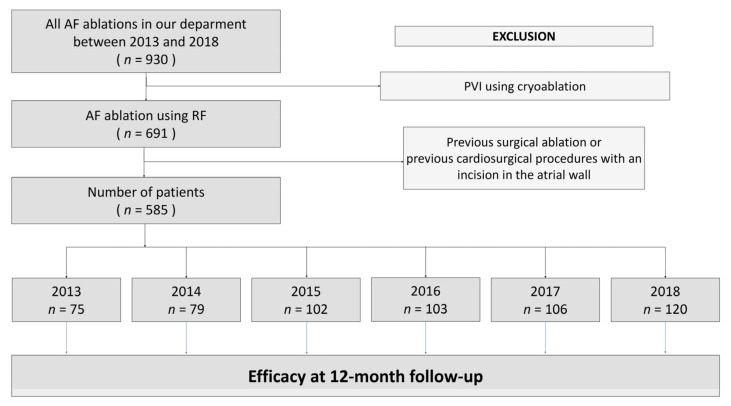
Study flow chart. Abbreviation: AF—atrial fibrillation, PVI—pulmonary vein isolation, RF—radiofrequency energy.

**Figure 2 medicina-58-00187-f002:**
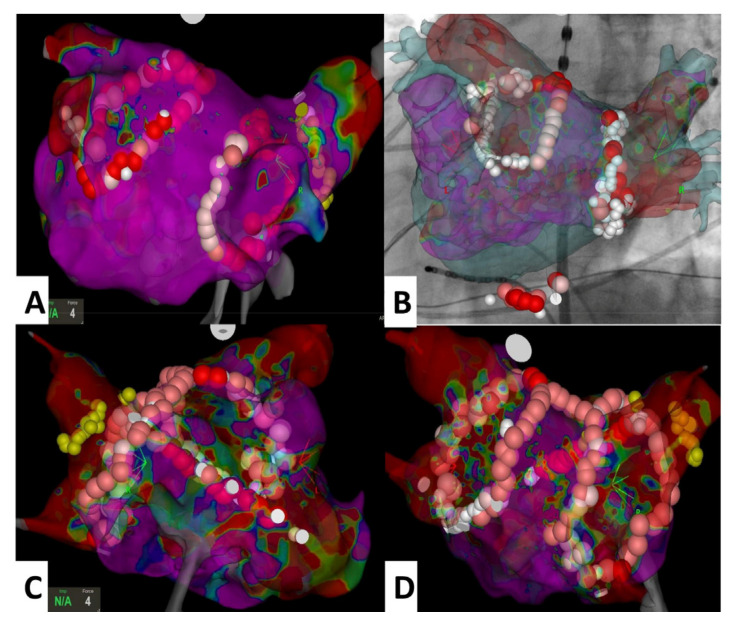
Views presenting the most common ablation lines performed for arrhythmogenic substrate modification. Panel (**A**) (PVI only), panel (**B**) (PVI and CTI line, Univu module), panel (**C**) and panel (**D**) (PVI, roof line and septal line). Abbreviation: CTI—cavotricuspid isthmus, PVI—pulmonary vein isolation.

**Figure 3 medicina-58-00187-f003:**
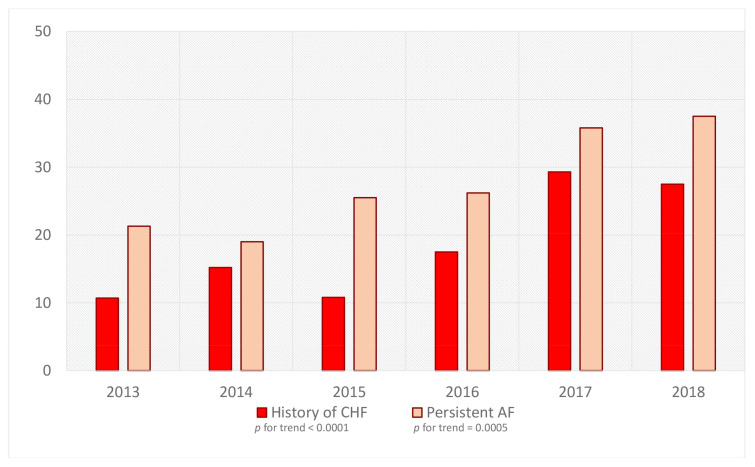
Percentage of patients with persistent AF and CHF burden undergoing ablation in our center in consecutive years. Abbreviations: AF—atrial fibrillation, CHF—chronic heart failure.

**Figure 4 medicina-58-00187-f004:**
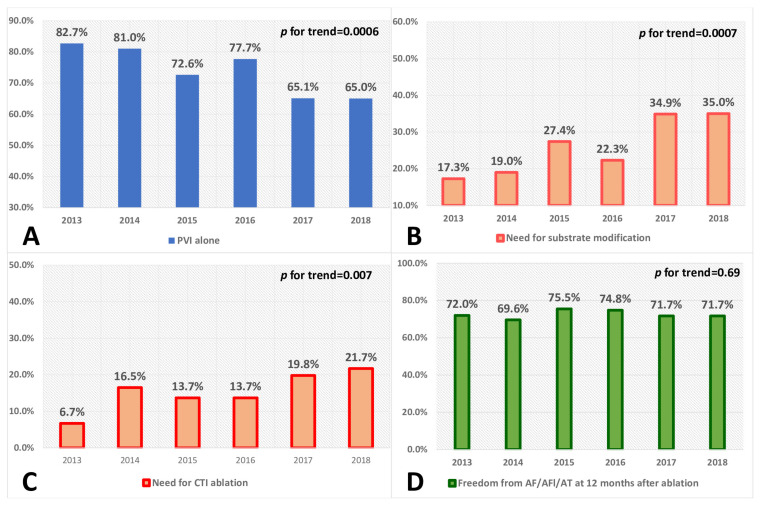
Percentage of patients who underwent AF ablation with PVI alone (Panel (**A**)), additional arrhythmogenic substrate modification (Panel (**B**)), and CTI ablation (Panel (**C**)); Panel (**D**) shows the percentage of patients free from AF/AFl/AT at the 12-month follow-up after ablation, regardless of the extent of the procedure in subsequent years. Abbreviations: PVI—pulmonary vein isolation, CTI—cavotricuspid isthmus, AF—atrial fibrillation, AFl—atrial flutter, AT—atrial tachycardia.

**Figure 5 medicina-58-00187-f005:**
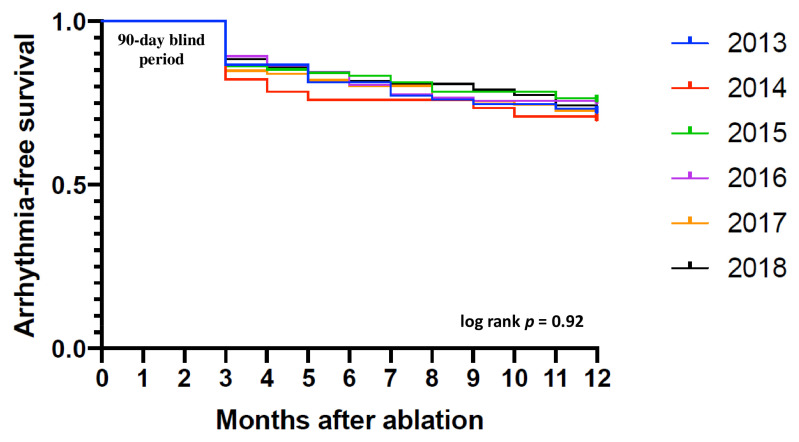
Kaplan–Meier curve showing time to the first recurrence of symptomatic or asymptomatic AF/AFl/AT up to the 12-month follow-up after ablation in consecutive years. Abbreviations: AF—atrial fibrillation, AFl—atrial flutter, AT—atrial tachycardia.

**Table 1 medicina-58-00187-t001:** Baseline characteristics.

	Total	2013	2014	2015	2016	2017	2018	*p* Value for Trend
Number of patients	585	75	79	102	103	106	120	
Sex (males), (%)	65.8	66.7	62.0	59.8	67.0	69.8	68.3	0.29
Age (years), mean (SD)	59 (10)	59 (10)	60 (10)	58 (11)	58 (10)	60 (11)	59 (12)	0.19
BMI (kg/m^2^), median (Q1; Q3)	29.4 (26.3; 32.5)	30.0 (26.1; 33.8)	29.8 (27.7; 32.6)	29.9 (25.8; 32.9)	28.9 (26.1; 32.0)	29.0 (26.4; 32.0)	29.4 (26.4; 32.3)	0.06
CHADS-VASc, median (Q1; Q3)	2 (1; 3)	2 (1; 3)	1 (1; 3)	2 (1; 3)	2 (1; 3)	2 (1; 3)	2 (1; 3)	0.01
HASBLED, median (Q1; Q3)	1 (1; 2)	1 (1; 2)	1 (1; 2)	1 (1; 2)	1 (0; 2)	1 (1; 2)	1 (0; 2)	0.42
Hypertension, (%)	67.4	69.3	70.9	68.6	70.9	65.1	61.7	0.15
History of diabetes, (%)	23.6	32.0	30.4	24.5	19.4	21.7	18.3	<0.01
History of CAD, (%)	27.9	28.0	22.8	30.4	22.3	33.0	29.2	0.45
Hyperlipidemia, (%)	78.0	82.7	82.3	74.5	77.7	80.2	73.3	0.18
Previous MI, (%)	6.8	8.0	7.6	2.9	5.8	9.4	7.5	0.66
History of CHF, (%)	19.3	10.7	15.2	10.8	17.5	29.3	27.5	<0.001
Implanted ICD/CRT, (%)	5.6	4.0	3.8	2.0	2.9	6.6	11.7	<0.01
History of stroke, (%)	4.4	4.0	2.5	3.9	3.9	7.6	4.2	0.39
History of TIA, (%)	3.6	6.7	1.3	3.9	4.8	3.8	1.7	0.29
Prior AF ablation, (%)	25.1	16.0	20.3	17.7	22.3	31.1	37.5	<0.001
Atrial fibrillation								
Paroxysmal, (%)	71.5	78.7	81.0	74.5	73.8	64.2	62.5	<0.001
Persistent, (%)	28.5	21.3	19.0	25.5	26.2	35.8	37.5	<0.001

Abbreviations: BMI—body mass index, CAD—coronary artery disease, MI—myocardial infarction, CHF—chronic heart failure, ICD—implanted cardioverter-defibrillator, CRT—cardiac resynchronization therapy, TIA—transient ischemic attack.

**Table 2 medicina-58-00187-t002:** In-hospital and periprocedural data.

	Total	2013	2014	2015	2016	2017	2018	*p* Value for Trend
Number of patients	585	75	79	102	103	106	120	
Echocardiographic parameters								
LA diameter (mm), mean (SD)	43 (5)	42 (5)	42 (4)	42 (5)	43 (5)	44 (5)	44 (6)	<0.001
LV ESV (ml), mean (SD)	54 (25)	52 (19)	50 (16)	51 (23)	51 (26)	59 (28)	58 (28)	0.03
LV EDV (ml), mean (SD)	108 (34)	103 (29)	106 (28)	106 (32)	103 (34)	116 (36)	113 (37)	0.046
LV EF (%), mean (SD)	51 (9)	53 (6)	53 (6)	52 (9)	52 (8)	50 (10)	49 (10)	<0.001
Procedural data								
PVI alone, (%)	73.0	82.7	81.0	72.6	77.7	65.1	65.0	<0.001
LA substrate modification, (%)	27.0	17.3	19.0	27.4	22.3	34.9	35.0	<0.001
CTI ablation, (%)	16.6	6.7	16.5	13.7	17.5	19.8	21.7	<0.01
In-hospital data								
In-hospital recurrence of AF, (%)	11.8	19.7	16.5	10.8	8.7	9.4	10.0	0.03
Major complications, *n* (%)	7 (1.2)	1 (1.3)	0 (0.0)	2 (2.0)	1 (1.0)	1 (0.9)	2 (1.7)	0.73
Minor complications, *n* (%)	23 (3.9)	3 (4.0)	4 (5.0)	3 (2.9)	3 (2.9)	4 (3.8)	6 (5.0)	0.41
*Drugs at discharge*								
Beta-blockers, (%)	88.4	86.7	84.8	86.3	87.4	90.6	92.5	0.07
Propafenone, (%)	52.1	46.7	59.5	58.8	64.1	43.4	42.5	0.06
Ca-blockers, (%)	17.4	10.7	16.5	17.7	19.4	17.0	20.1	0.12
Sotalol, (%)	8.4	10.7	10.1	9.8	8.7	7.6	5.0	0.1
Amiodarone, (%)	8.4	6.7	11.4	8.8	4.9	8.5	10.0	0.81
VKA, (%)	52.7	74.7	84.8	68.6	44.7	35.8	25.8	<0.001
NOAC, (%)	47.3	25.3	15.2	31.4	55.3	64.2	74.2	<0.001

Abbreviations: LA—left atrium, LV—left ventricle, ESV—end-systolic volume, EDV—end-diastolic volume, EF—ejection fraction, PVI—pulmonary vein isolation, CTI—cavotricuspid isthmus, AF—atrial fibrillation, Ca-blocker—calcium channel blockers, VKA—vitamin K antagonist, NOAC—new oral anticoagulant.

**Table 3 medicina-58-00187-t003:** Follow-up data showing freedom from AF at 6, 9, and 12 months after ablation.

	Total	2013	2014	2015	2016	2017	2018	*p* Value for Trend
Freedom from AF/AFl/AT at 6 months following ablation, (%)	80.7	81.3	76.0	83.3	80.6	80.2	81.7	0.75
Freedom from AF/AFl/AT at 9 months following ablation, (%)	76.4	74.7	73.4	78.4	75.7	75.5	79.2	0.48
Freedom from AF/AFl/AT at 12 months following ablation, (%)	72.6	72.0	69.6	75.5	74.8	71.7	71.7	0.69

Abbreviation: AF—atrial fibrillation, AFl—atrial flutter, AT—atrial tachycardia.

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
