# Peer review of "Temporary Trends Concerning the Extent and Efficacy of Atrial Fibrillation Ablation Using Radiofrequency Energy in a Polish Single-Center Experience"

_medicina, 2022, doi:10.3390/medicina58020187_

Round 1

Reviewer 1 Report

The authors have drafted a rather thorough report on their single-center experience using radiofrequency for catheter atrial fibrillation ablation over a 5-year period.  Though the literature is already replete with these single-center experiences/studies, this study is unique in that it occurred in a primarily Eastern European / Polish population.  This uniqueness needs to be emphasized and the results elaborated in this context.  The trend of peri-procedural characteristics changing over the 5 years is also intriguing and warrants further discussion.   A few minor and major points, if addressed, would make this manuscript stronger (see below)

Minor Comments

Line 42

I would not use the word "invasive" here.  Ablation used to occur surgically during open heart surgery, which was very invasive, so ablation transvenously is relatively non-invasive.  I would just say patients "who received catheter ablation in our department"

Line 99

Take out "a" before normal and "a" before diseased

Line 105

Even catheter manipulation can induce typical flutter.  Simply say typical flutter that was induced during the study or documented beforehand were treated with a CTI line.

Line 121

Need "an" before atrioesophageal fistula

Line 122

"Endoscopic diagnosis" can be shortened to "endoscopy"

Line 126

peri-procedural should be peri-procedurally

Figure 4:  Y axis percentages can be made more visible; numbers on top of bars can also be enlarged

Line 203

I would not call this incomplete control; recurrence of atrial arrhythmia should suffice

Line 227

Again, i would not use "invasive" here 

Major Comments

  1. One unique feature of this study is the population (Polish/Eastern European) involved in this study. How does this population compare to populations in other large AF ablation studies?  Please comment with respect to risk factors of AF and recurrence of AF after ablation (incidence of sleep apnea, obesity, diabetes, and HTN) for this patient population.  A good place to mention this would be in the introduction section.

  1. The trend toward performing ablations in patients with greater severity of AF, worse LV function, and increasing CHAD2SVASC score needs to be explored further. Were referred patients simply more ill?  Did the operators become more comfortable doing cases of AF that were more severe?  Or did clinical guidelines simply impact practice patterns?  

  1. Line 310

The authors identify that one of the limitations in this study was followup conducted only via intermittent ECGs and Holter monitoring, which is the standard of care.  However, studies where continuous long-term monitoring was employed using CIED for followup invariably demonstrated a higher rate of atrial arrhythmia.  The true recurrence rate likely being higher should be explicitly stated here. 

  1. It would be interesting to see what percentage of these patients received antiarrhythmics after the 3 month blanking period

  1. Is there data on procedure time and total ablation time? A trend of procedure time and ablation time of the years would be useful. 

  1. Percentage of patients ablated using CARTO vs ESI over the years would also be useful, can be included in Table 2

Author Response

At the beginning I would like to thank you for your constructive comments.

Minor Comments

Comment 1: Line 42

I would not use the word "invasive" here.  Ablation used to occur surgically during open heart surgery, which was very invasive, so ablation transvenously is relatively non-invasive.  I would just say patients "who received catheter ablation in our department"

Response: The sentence has been reedited.

Comment 2: Line 99

Take out "a" before normal and "a" before diseased

Response: The sentence has been removed.

Comment 3: Line 105

Even catheter manipulation can induce typical flutter. Simply say typical flutter that was induced during the study or documented beforehand were treated with a CTI line.

Response: The sentence has been replaced.

Comment 4: Line 121

Need "an" before atrioesophageal fistula

Response: The phrase has been introduced.

Comment 5: Line 122

"Endoscopic diagnosis" can be shortened to "endoscopy"

Response: The sentence has been corrected.

Comment 6: Line 126

peri-procedural should be peri-procedurally

Response: The phrase has been introduced.

Comment 7: Figure 4: Y axis percentages can be made more visible; numbers on top of bars can also be enlarged.

Response:   The Figure 4 has been reedited.

Comment 8: Line 203

I would not call this incomplete control; recurrence of atrial arrhythmia should suffice.

Response:   The sentence has been reedited.

Comment 9: Again, i would not use "invasive" here.

Response: The sentence has been removed.

Major Comments

Comment 1: One unique feature of this study is the population (Polish/Eastern European) involved in this study. How does this population compare to populations in other large AF ablation studies?  Please comment with respect to risk factors of AF and recurrence of AF after ablation (incidence of sleep apnea, obesity, diabetes, and HTN) for this patient population.  A good place to mention this would be in the introduction section.

Response:   These comments were introduced in the re-edited introduction section.

Comment 2: The trend toward performing ablations in patients with greater severity of AF, worse LV function, and increasing CHAD2SVASC score needs to be explored further. Were referred patients simply more ill?  Did the operators become more comfortable doing cases of AF that were more severe?  Or did clinical guidelines simply impact practice patterns?  

Response: These comments were included in the redrafted discussion section.

Comment 3: Line 310

The authors identify that one of the limitations in this study was followup conducted only via intermittent ECGs and Holter monitoring, which is the standard of care.  However, studies where continuous long-term monitoring was employed using CIED for followup invariably demonstrated a higher rate of atrial arrhythmia.  The true recurrence rate likely being higher should be explicitly stated here. 

Response:   Thank you for this comment. In this population, only a small group 13.5% (79/585) was also monitored with implantable devices to assess the efficacy of ablation. We also agree that the true recurrence rate is likely being higher, as we highlighted in the text.

Comment 4: It would be interesting to see what percentage of these patients received antiarrhythmics after the 3 month blanking period

Response: Unfortunately, we don’t have this data.

Comment 5: Is there data on procedure time and total ablation time? A trend of procedure time and ablation time of the years would be useful.

Response: Unfortunately, we don’t have this data.

Comment 6:  Percentage of patients ablated using CARTO vs ESI over the years would also be useful, can be included in Table 2

Response:   In the collected group of patients, only 23 patients underwent ablation with the Ensite system. Therefore, we didn’t include these data in the re-edited manuscript.

Reviewer 2 Report

See attached file

Referee report

Article ID: medicina-1547233

Title: Temporary trends concerning the extent and efficacy of atrial 2 fibrillation ablation using radiofrequency energy in a Polish 3 single-center experience..

General Comments

The main objective of this study was to analyze the temporary trends in the baseline characteristics, echocardiographic and periprocedural parameters, including the extent of a single AF ablation procedure using radiofrequency energy (RF) between 2013 and 2018 in a single-center experience. The efficacy of the procedure underwent assessment during 12-month follow-up. I think that this kind of paper is important to Medicina readers, however, we ask you a series of questions and suggestions throughout the text. So, after to respond favourably to all these questions and suggestions, in my opinion, the paper can be accepted for publication. I suggest expanding the introduction section with the text used in the discussion section. I suggest merging the text from the results section into the discussion section.

Some specific comments:

Introduction

  1. Page 1. You should expand the introduction section, adding epidemiological support to justify your work.  

Materials and Methods:

  1. Page 2. Lines 45-49. Move the text of the objective to the end of the introduction section.
  2. Page 2. Line 50. What is the research design used in this article?. This is indicated at the beginning of the discussion section: This was a single-centre retrospective cohort study evaluating….

Results:

For a better understanding of your tables and figures, please present each one, similar to the text used in figure 1: “The group we analyzed included 585 patients, mostly with paroxysmal AF (71.5%). 163 The study flow chart for the analysis is shown in Figure 1”. You can use texts such as:

  • Table X shows….
  • Fig. X illustrates….
  • Table X presents….
  • Fig. X summarize….
  •  

Modify this part of your article. Transfer the text describing the results of the table to the Discussion section. In addition, where the results are commented is in the discussion section.

Examples:

Over the years, patients with increasingly larger LA dimensions (p<0.0001) and lower LVEF (p=0.0003) were qualified for invasive treatment. The extent of the procedure in-volving only PVI (73% of all patients) decreased in subsequent years and reached 65% in 2018. At the same time, the percentage of patients who underwent CTI ablation (p=0.007) and arrhythmogenic substrate modification increased from 17.3% in 2013 to 35% in 2018 (p=0.0007) (Figure 4). In the hospital setting, the rate of recurrent arrhythmias decreased (p=0.03); the percentages of procedure-related (minor and major) complications remained 186 comparable over the years. Among antiarrhythmic drugs at hospital discharge, beta-blockers (88.4%) and propafenone (52.1%) were the most common, with a low percentage of amiodarone use (8.4%). This trend was observed in subsequent years. In 2015/2016, NOACs became the predominant group of drugs used for thromboprophylaxis - 25.3% of 190 patients in 2013 compared to 74.2% in 2018 (p<0.0001) (Table 2).

Discussion:

Not only discuss the benefits of these electrophysiology procedures, but also raise the dangers of radiation to patients and operators.

Order the writing of this section. You must necessarily comment or compare your results (all tables and figures) inside or with other articles.

Author Response

At the beginning I would like to thank you for your constructive comments.

Comment 1: 

Introduction. Page 1. You should expand the introduction section, adding epidemiological support to justify your work.  

Response:   These comments were introduced in the re-edited introduction to the manuscript.

Comment 2: 

Materials and Methods: Page 2. Lines 45-49. Move the text of the objective to the end of the introduction section.

Response: The text has been moved.

Comment 3: 

Materials and Methods: Page 2. Line 50. What is the research design used in this article?. This is indicated at the beginning of the discussion section: This was a single-centre retrospective cohort study evaluating.

Response: The data has been completed

Comment 4: 

Results: For a better understanding of your tables and figures, please present each one, similar to the text used in figure 1: “The group we analyzed included 585 patients, mostly with paroxysmal AF (71.5%). 163 The study flow chart for the analysis is shown in Figure 1”. You can use texts such as: Table X shows…., Fig. X illustrates…., Table X presents…., Fig. X summarize….,

Modify this part of your article. Transfer the text describing the results of the table to the Discussion section. In addition, where the results are commented is in the discussion section.

Response:   These comments have been incorporated into the re-edited manuscript.

Comment 5: 

Discussion: Not only discuss the benefits of these electrophysiology procedures, but also raise the dangers of radiation to patients and operators.

Response:   We don’t have accurate radiation exposure data, but we comment the dangers of radiation to patients and operators in the re-edited discussion seection.

Comment 6: 

Discussion: Order the writing of this section. You must necessarily comment or compare your results (all tables and figures) inside or with other articles.

Response:   These comments have been incorporated into the re-edited manuscript.

Round 2

Reviewer 2 Report

The authors answer all the questions correctly and take into account the suggested changes. Finally, in my opinion, the paper must be accepted for publication.